# Balloon-Shaped SMF Blood Glucose Concentration and Temperature Sensor Based on Core-Offset Structure

**DOI:** 10.3390/s24196344

**Published:** 2024-09-30

**Authors:** Weihua Zhang, Yixi Liu, Zhengrong Tong, Xue Wang, Yipeng Tao, Haozheng Yu, Jinlin Mu

**Affiliations:** Engineering Research Center of Optoelectronic Devices and Communication Technology, Ministry of Education, Tianjin Key Laboratory of Film Electronic and Communication Devices, School of Integrated Circuit Science and Engineering, Tianjin University of Technology, Tianjin 300384, China; nmgzwh@163.com (W.Z.);

**Keywords:** blood glucose concentration sensor, Mach–Zehnder interferometer (MZI), balloon-shaped single-mode fiber (SMF), dual-parameter measurement

## Abstract

A blood glucose concentration and temperature sensor with a balloon-shaped single-mode fiber (SMF) based on a core-offset structure is proposed and experimentally demonstrated. The balloon-shaped SMF is created by offset-fusing a straight-line SMF between two other SMFs, thereby forming a Mach–Zehnder interferometer (MZI). The core-offset structure can effectively excite higher-order cladding modes. The experimental results showed that the maximum sensitivity of blood glucose concentration was 0.331 nm/(mmol/l) and the maximum sensitivity of temperature was 0.216 nm/°C when the offset distance was 10 μm. Dual-parameter measurement was achieved through a dual-parameter matrix. In addition, the sensor has characteristics such as simple structure, low cost, good stability, and electromagnetic interference resistance, making it potentially valuable for diagnosing high blood glucose and related conditions.

## 1. Introduction

Diabetes is a metabolic disease characterized by high blood glucose [1], which has become a chronic disease and poses a major threat to human health. Diabetes is a disease caused by insufficient production of insulin and/or inadequate utilization of insulin [2], which leads to elevated blood glucose levels. The disability and mortality rates of diabetes are very high [3]. With the increasing prevalence of diabetes, timely and effective blood glucose monitoring is becoming important for diabetes patients [4]. Effective blood glucose monitoring can help patients control their blood glucose levels, adjust their medication and lifestyle accordingly, and avoid long-term complications of the disease. It is recommended that diabetes patients regularly monitor their blood sugar levels to ensure good control of their disease. Therefore, accurate and appropriate blood glucose detection technology has important applications and has become a hot research topic [5].

In recent years, various types of blood glucose concentration sensors have been studied, such as optical sensors [6], electrode method [7], chemiluminescence method, and electrochemical sensors [8]. Due to the anti-electromagnetic interference, fast response, and chemical inertness of fiber optic sensors, fiber optic blood glucose sensors have been widely researched [9]. In 2015, Lokman et al. [10] proposed a simple inline Mach–Zehnder interferometer (MZI) based on a dumbbell-shaped structure for detecting glucose concentrations. In 2019, Lidiya et al. [11] proposed a D-shaped photonic crystal fiber doped with gold nanoparticles for measuring blood glucose levels. Due to the high cost of the sensor, it is difficult to use in clinical detection. In 2022, Zhong et al. [12] proposed a helical intermediate-period fiber grating glucose sensor. In the test of the glucose concentration, ranging from 0.1 mg/mL to 200 mg/mL, the sensitivity of the sensor was 0.026 nm/(mg/mL). The temperature characteristic was studied, and the sensor had a temperature sensitivity of 3.76 pm/°C from 30 °C to 100 °C. The sensor has low glucose sensitivity and crosstalk of temperature. In 2023, Cheng et al. [13] proposed a label-free glucose biosensor based on a tapered two-mode fiber sandwiched between two single-mode fibers. The measurement range was 0 mg/mL to 270 mg/mL and the maximum glucose sensitivity was 0.112 nm/(mg/mL). In 2023, Li et al. [14] proposed a waveguide Bragg grating sensor for blood glucose monitoring. By exploiting glucose oxidase as the upper cladding, polydimethylsiloxane was used as the substrate, and polymethyl methacrylate was used as the core layer. In the blood glucose concentration range of 0 mg/mL to 3.6 mg/mL, the blood glucose sensitivity of the sensor was 242.9 pm/(mg/mL). However, the blood glucose concentration sensitivity of fiber grating is generally low. Methods to sensitize fiber grating generally increase the cost of the sensor and the complexity of the manufacturing process [15].

Therefore, a balloon-shaped SMF blood glucose concentration sensor based on the core-offset structure is proposed here. The core-offset structure can effectively excite the higher-order cladding modes [16]. Modal interference is formed due to the optical path difference between different modes. The sensor has a high sensitivity to blood glucose concentration. The sensitivities of the interference valley formed by different interference modes to blood glucose concentration and temperature are different. Dual-parameter measurement of glucose concentration and temperature is implemented by the dual-parameter matrix. The proposed sensor has high sensitivity, simple structure, and low cost with a short response time. It holds significant potential application value in the fields of medical diagnosis and physiological health, particularly in diagnosing high blood glucose and related conditions.

## 2. Materials and Methods

### 2.1. Balloon-Shaped SMF Fabrication

The structure diagram of the balloon-shaped SMF blood glucose and temperature sensor based on the core-offset structure is shown in Figure 1. The higher-order cladding modes are excited by bending the straight SMF into balloon-shaped SMF. In Figure 1, d is the bending diameter of the balloon-shaped SMF, and l is the core-offset distance. To improve the sensitivity of the sensor, a core-offset structure was used to construct MZI in the balloon-shaped SMF. When light enters the core-offset structure from the incident SMF, the light in the core leaks into the cladding. The leakage of light energy depends on the size of the offset. When the light propagates to the output core-offset structure, part of the light in the cladding is coupled to the core. Due to different propagation paths, the phase difference results in modal interference.

The SMF (YOFC/SMF-28) was selected for the experiment. The core diameter of SMF is 8.2 μm and the cladding diameter is 125 μm. The core refractive index is 1.4502 and the cladding refractive index is 1.445. The manufacturing process of the sensor is shown in Figure 2. In the first step, the input SMF and the sensing SMF were fused into a core-offset structure with a fusion splicer (S187C, Sumitomo Corporation, Kansai, Japan). In the second step, the output SMF was fused by the same core-offset distance. In the last step, the input SMF and output SMF were inserted into a capillary tube with a diameter of 0.4 mm to form a balloon shape. The bending diameter of the balloon-shaped SMF was adjusted by a capillary tube.

### 2.2. Theoretical Principle

The actual bending interference area is only a portion of the bending probe; therefore, the measured bending diameter is slightly larger than the actual value. Due to the stress-optical effect [13], when SMF is bent, the RI distribution across the cross-section is no longer symmetric. The refractive index of SMF increased on the tensile side and decreased on the compressive side. The refractive index distribution of the conformal mapping of bent SMF into the equivalent straight SMF is shown in Figure 3. After conformal mapping, the bent SMF refractive index is defined as [17]:(1)$$n_{bent}=n_{1}\left(1+\frac{x}{d_{eff}}\right)$$
where *n*_1_ is the refractive index distribution of the straight SMF, and *x* is the distance from the core center. The outside of the bent part is positive, and the inside is negative. *d_eff_* is the bending diameter after equivalence, which can be expressed as [18]:(2)$$d_{eff}=\frac{d}{1-n_{1}^{2}\left[P_{12}-\upsilon (P_{11}+P_{12})\right]/2}$$
where *P*_12_ and *P*_11_ are components of the photo-elastic tensor, and υ is Poisson’s ratio. For quartz fiber, $d_{eff}\approx 1.28\cdot d$. Therefore, the equivalent refractive index of the bent SMF is:(3)$$n_{bent}=n_{1}\left(1+\frac{x}{1.28d}\right)$$

When light enters the core-offset structure from the incident SMF, high-order cladding modes are excited. Due to the different propagation paths, the phase difference is generated between the core mode and the cladding modes, resulting in modal interference at the core-offset structure. The wavelength of the interference valley can be represented as:(4)$$\lambda =\frac{2\left(\Delta n_{core,eff}-\Delta n_{clad,eff}\right)L}{2M+1}$$
where $\Delta n_{core,eff}$ is the difference in the effective refractive index of the core and $\Delta n_{core,eff}$ is the difference in the effective refractive index of the cladding, M=0,1,2,3……, and L is the effective length of MZI, $L=\frac{1}{2}\pi d$.

The effective refractive index of the cladding mode is changed with the variation in the environmental parameters. The core of the fiber is not directly exposed to the external environment, so the effective refractive index of the core remains unchanged. Therefore, the wavelength of the interference valley shifts with the change in environmental parameters. When the external blood glucose concentration changes, the wavelength change in the interference valley can be expressed as:(5)$$\Delta \lambda_{C}=\frac{\partial \lambda }{\partial C}=\frac{\lambda }{\Delta n_{eff}}\cdot \frac{\partial \Delta n_{eff}}{\partial C}$$
where $\Delta n_{eff}$ is the difference in the effective refractive index between core mode and cladding mode, $\Delta n_{eff}=\Delta n_{core,eff}-\Delta n_{clad,eff}$. When the environment temperature changes, the wavelength change in the interference valley can be expressed as [19]:(6)$$\Delta \lambda_{T}=\frac{\partial \lambda }{\partial T}=\lambda \left(\frac{dL}{LdT}+\frac{d\Delta n_{eff}}{\Delta n_{eff}dT}\right)=\lambda \left(\alpha +\xi \right)$$
where α is the thermal expansion coefficient of SMF, and ξ is the thermo-optical coefficient of SMF. When the environment temperature changes, the thermal expansion coefficient and thermal optical coefficient of the SMF change. Therefore, the wavelength of the interference valley also changes linearly.

Before the experiment, the light energy transfer spectrum of the balloon-shaped SMF based on a core-offset structure was simulated using the beam PROP method in Rsoft, where the background refractive index was 1 and the central wavelength was 1550 nm. The simulation length of the input SMF and output SMF was set to 1 mm, the bending diameter was set to 1.5 cm, and the offset was set to 10 μm. Figure 4 shows the energy field distribution diagram and normalized energy diagram. As can be seen from the simulation results, when the light was transmitted to the core-offset structure, the light energy in the core leaked into the cladding, and modal interference occurred due to the different propagation paths of the light.

### 2.3. Experimental Device

The blood glucose concentration sensing experimental device is shown in Figure 5. It includes supercontinuum sources (SC-5, Anyang Photonics, Wuhan, China) with a wavelength range of 470 nm to 2400 nm and spectral stability (800–1700 nm) < 0.1 dB, an optical spectrum analyzer (OSA, AQ6370D, Yokogawa, Tokyo, Japan) with a resolution of 0.02 nm and spectrum analysis range of 600 nm~1700 nm, a temperature controller (GDS-50, Test, Nanjing, China) with temperature fluctuation of ±0.5 °C, and a brix-salt meter (PAL-BX|SALT, ATAGO, Tokyo, Japan) with the brix detection range from 0% to 90% and resolution of 0.1%. During the experiment of the blood glucose concentration, we cleaned the sensor with pure water to eliminate the influence of residual blood on the next set of experiments after each test. In the temperature experiment, the environment temperature was controlled by the temperature controller. The sensor was immersed in a small amount of blood to achieve the effect of uniform heating. Spectra were recorded whenever the environment temperature rose by 5 °C.

## 3. Results

### 3.1. Morphology Analysis of the Sensor

Three sensors with different parameters were manufactured, and their bending diameters and offsets are shown in Table 1. Figure 6 shows the spatial spectrum diagram of l = 10 μm. It can be seen that the energy intensity of the primary cladding mode was significantly higher than that of the weak cladding mode, which confirmed the existence of modal interference. Ignoring the interference from the weak cladding mode, the two main peaks in the figure corresponded to the interference of the first and second cladding modes, respectively. The responses of the two interference modes to blood glucose concentration and temperature were different. Therefore, the sensitivity of the interference valley generated by the interference of different cladding modes to blood glucose concentration and temperature were different. The dual-parameter measurement was achieved through the dual-parameter matrix. The changes in blood glucose concentration and temperature can be represented as a matrix:(7)$$\left[\begin{array}{l} \Delta \lambda_{dipX}\\ \Delta \lambda_{dipY} \end{array}\right]=\left[\begin{array}{cc} K_{dipX}^{C} & K_{dipX}^{T}\\ K_{dipY}^{C} & K_{dipY}^{T} \end{array}\right]\left[\begin{array}{c} \Delta C\\ \Delta T \end{array}\right]$$
where $K_{dipX}^{C}$ and $K_{dipX}^{T}$ are the blood glucose concentration sensitivity and temperature sensitivity of dipX, respectively, while $K_{dipY}^{C}$ and $K_{dipY}^{T}$ are the blood glucose concentration sensitivity and temperature sensitivity of dipY, respectively.

### 3.2. Experimental Results

#### 3.2.1. Blood Glucose Concentration Experiment Results

In the blood glucose concentration experiment, the blood glucose concentration was measured from 1.1 mmol/L to 34.1 mmol/L. Sensors 1~3 were red-shifted in this blood glucose concentration range. Figure 7 shows the experimental results of sensor 1 for blood glucose concentration, with a maximum blood glucose concentration sensitivity of 0.081 nm/(mmol/L). Figure 8 shows the experimental results of sensor 2 for blood glucose concentration. Sensor 2 had a maximum sensitivity of 0.187 nm/(mmol/L) in the range of blood glucose concentration from 1.1 mmol/L to 34.1 mmol/L. Figure 9 shows the experimental results of sensor 3 for blood glucose concentration. The blood glucose concentration sensitivities of dips 1~4 were 0.226 nm/(mmol/L), 0.234 nm/(mmol/L), 0.248 nm/(mmol/L), and 0.331 nm/(mmol/L), respectively. The maximum sensitivity of blood glucose concentration for sensors with different offsets is compared in Figure 10. It can be seen that the core-offset structure is an effective way to increase the sensitivity of blood glucose concentration.

#### 3.2.2. Temperature Experiment Results

In the temperature experiment, the measuring range of temperature was from 5 °C to 45 °C. When the temperature was stabilized for 10 min, we recorded the spectrum. Sensors 1~3 were red-shifted in this temperature range. Figure 11 shows the temperature experiment results of sensor 1, with a maximum blood glucose concentration sensitivity of 0.080 nm/°C. It can be seen that for the balloon-shaped SMF without the core-offset structure, the temperature sensitivity was low. Figure 12 shows the temperature experiment results of sensor 2, with a maximum temperature sensitivity of 0.169 nm/°C. Figure 13 shows the temperature experiment results of sensor 3. The temperature sensitivities of dips 1~4 were 0.216 nm/°C, 0.164 nm/°C, 0.104 nm/°C, and 0.183 nm/°C, respectively. The comparison of the maximum temperature sensitivity of sensors with different offsets is shown in Figure 14. As can be seen from the figure, sensors 1~3 had a high linear fitting degree. Therefore, the core-offset structure is an effective sensitization method for the balloon-shaped SMF.

According to the experimental results of sensor 3 and Equation (7), the relationship between wavelength drift of the interference valley and blood glucose concentration and temperature can be expressed as follows:(8)$$\left[\begin{array}{l} \Delta C\\ \Delta T \end{array}\right]={\left[\begin{array}{cc} 0.226\mathrm{nm}/(\mathrm{mmol}/l) & 0.216\mathrm{nm}/{}^{\circ}C\\ 0.331\mathrm{nm}/(\mathrm{mmol}/l) & 0.183\mathrm{nm}/{}^{\circ}C \end{array}\right]}^{-1}\left[\begin{array}{c} \Delta \lambda_{dip1}\\ \Delta \lambda_{dip4} \end{array}\right]$$

Therefore, a dual-parameter measurement of blood glucose concentration and temperature can be achieved using a balloon-shaped SMF sensor based on the core-offset structure.

Stability is also an important indicator for sensors, so the stability of the balloon-shaped SMF sensor based on the core-offset structure was studied. The sensor was placed in different blood glucose concentrations and different environmental temperatures. The spectra were recorded continuously for 120 min at ten-minute intervals. Figure 15 shows the stability experiment results of sensor 3, where the maximum wavelength fluctuations of dip1 in 1.1 mmol/L and 34.1 mmol/L were 0.07 nm and 0.12 nm, respectively. The maximum wavelength fluctuations of dip4 in 1.1 mmol/L and 34.1 mmol/L were 0.08 nm and 0.17 nm, respectively. The maximum wavelength fluctuations of dip1 at 5 °C and 45 °C were 0.10 nm and 0.11 nm, respectively, and the maximum wavelength fluctuations of dip4 at 5 °C and 45 °C were 0.09 nm and 0.13 nm, respectively. Therefore, the balloon-shaped SMF sensor based on the core-offset structure had high stability.

Comparing the stability with other sensors, the comparison results are shown in Table 2. High sensitivity can also lead to lower stability. As we made improvements to the sensor’s structure, we refrained from modifying the sensor’s surface (such as coating with sensitizing materials and surface plasmon resonance). This was chosen to strike a balance between sensitivity and stability. Therefore, the stability of the sensor we proposed was better. 

## 4. Conclusions

In summary, a balloon-shaped SMF glucose concentration and temperature sensor based on the core-offset structure was proposed. The sensor uses a fusion splicer for core-offset fusion and then uses a capillary tube to form a balloon-shaped structure. The simulation analysis of the transmission spectrum and spatial spectral transformation of the balloon-shaped SMF sensor based on the core-offset structure were conducted to verify its feasibility. The experimental results showed that the maximum sensitivity of the sensors was 0.331 nm/(mmol/L) in the blood glucose concentration range from 1.1 mmol/L to 34.1 mmol/L, and the maximum sensitivity was 0.216 nm/°C in the temperature range of 5 °C to 45 °C. In addition, the sensors have characteristics such as simple structure, low cost, good stability, and electromagnetic interference resistance. Therefore, the balloon-shaped SMF blood glucose concentration and temperature sensor based on the core-offset structure has great potential in biomedical detection.

## Figures and Tables

**Figure 1 sensors-24-06344-f001:**
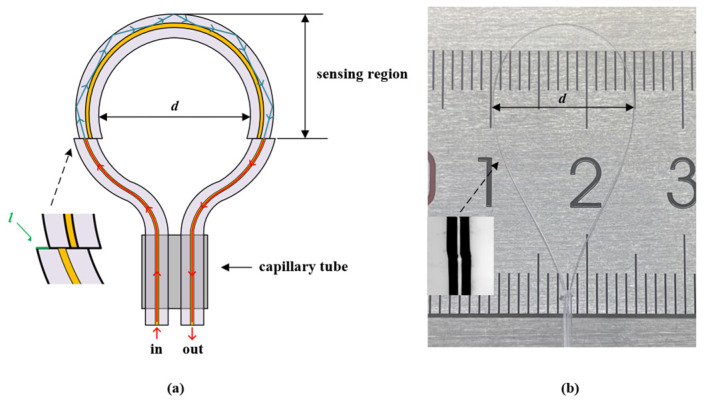
(**a**) The schematic diagram of the proposed sensor. (**b**) The physical diagram of the proposed sensor.

**Figure 2 sensors-24-06344-f002:**
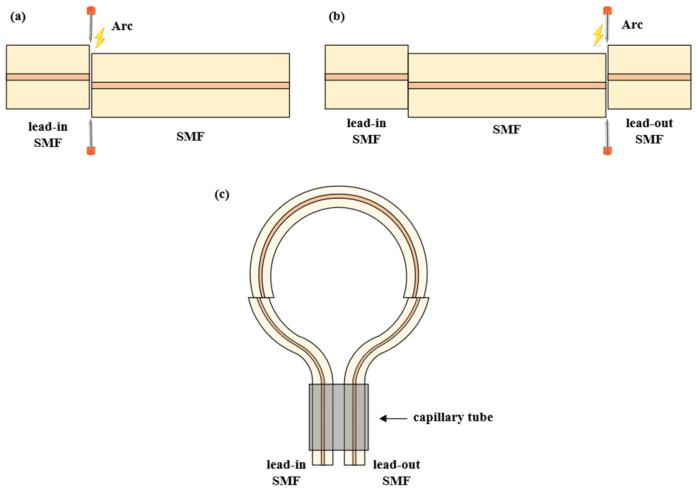
The fabrication process of the balloon-shaped SMF based on the core-offset structure. (**a**) First staggered-core structural fabrication. (**b**) Second staggered-core structure fabrication. (**c**) Fabrication of balloon structure.

**Figure 3 sensors-24-06344-f003:**
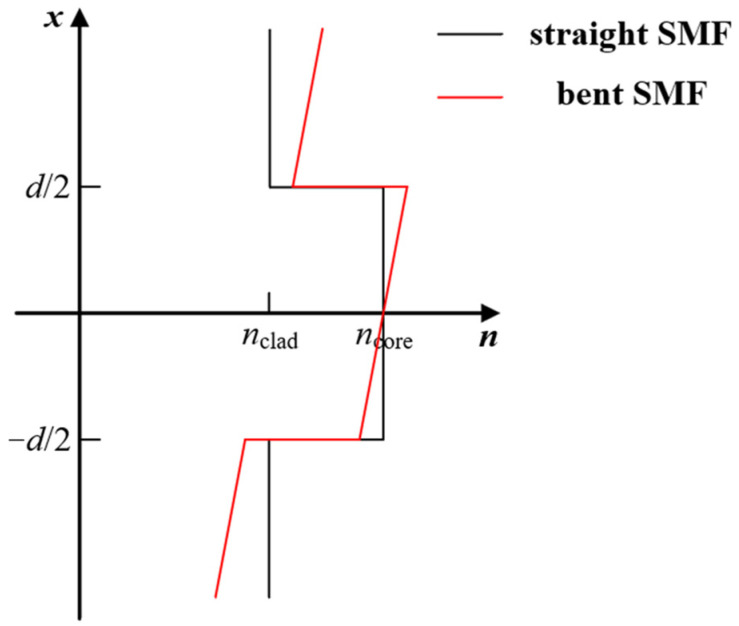
Refractive index distribution of SMF.

**Figure 4 sensors-24-06344-f004:**
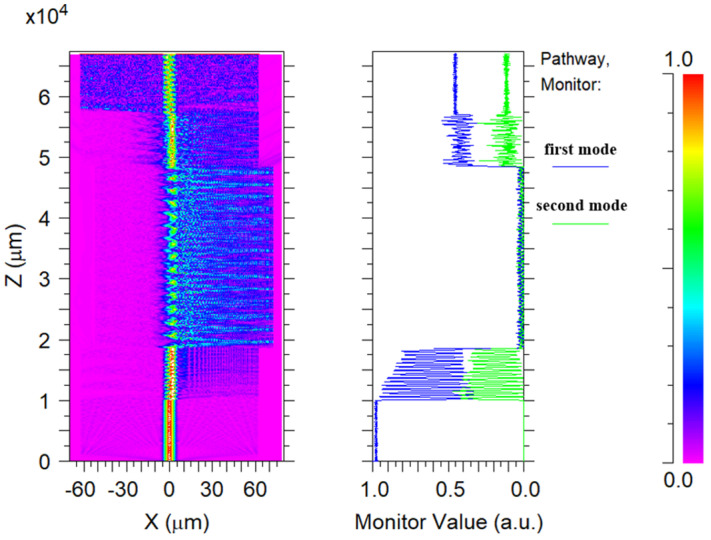
The energy field distribution diagram and normalized energy diagram of l = 10 μm.

**Figure 5 sensors-24-06344-f005:**
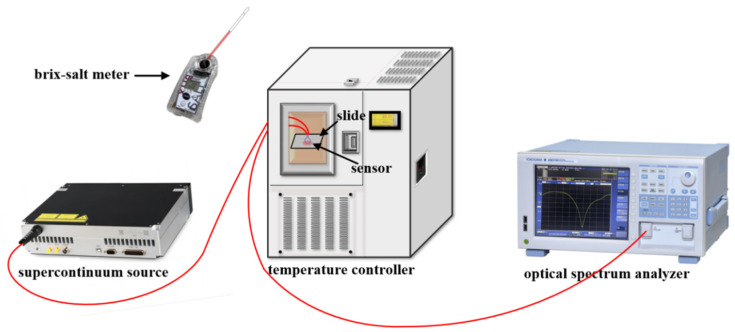
Experimental sensing device.

**Figure 6 sensors-24-06344-f006:**
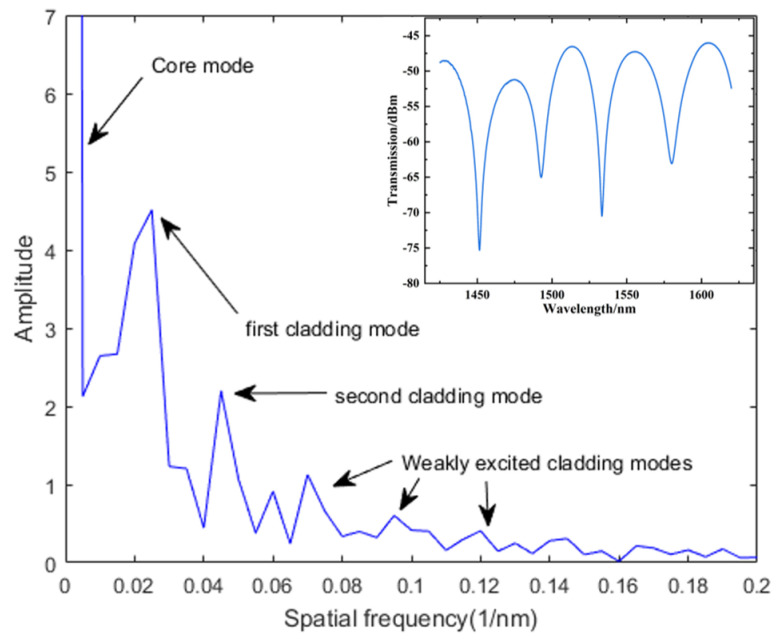
Spatial spectrum diagram of l = 10 μm.

**Figure 7 sensors-24-06344-f007:**
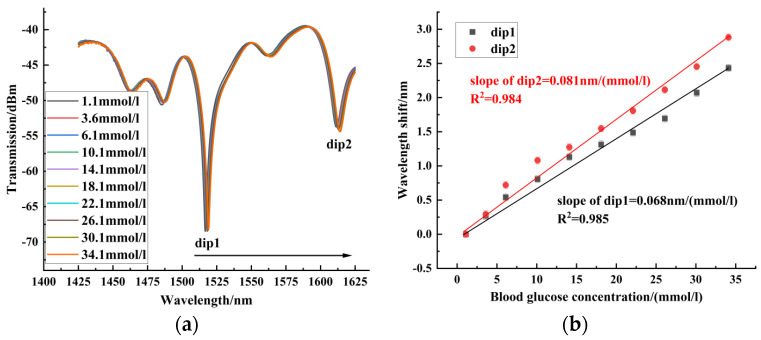
Blood glucose concentration experimental results of sensor 1. (**a**) Spectrogram of the blood glucose concentration experiment of sensor 1. (**b**) Experimental linear fit of blood glucose concentration for Sensor 1.

**Figure 8 sensors-24-06344-f008:**
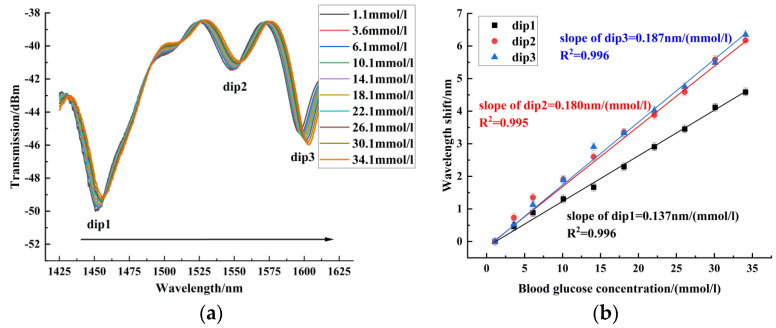
Blood glucose concentration experimental results of sensor 2. (**a**) Spectrogram of the blood glucose concentration experiment of sensor 2. (**b**) Experimental linear fit of blood glucose concentration for Sensor 2.

**Figure 9 sensors-24-06344-f009:**
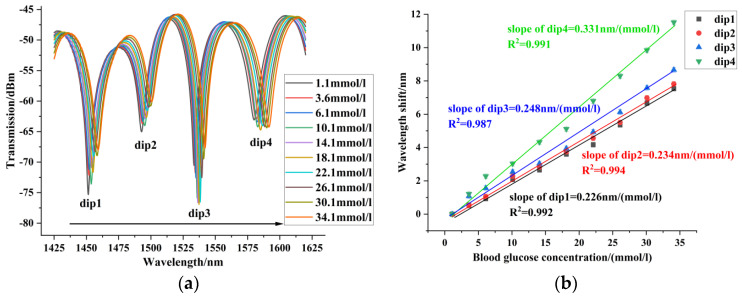
Blood glucose concentration experimental results of sensor 3. (**a**) Spectrogram of the blood glucose concentration experiment of sensor 3. (**b**) Experimental linear fit of blood glucose concentration for Sensor 3.

**Figure 10 sensors-24-06344-f010:**
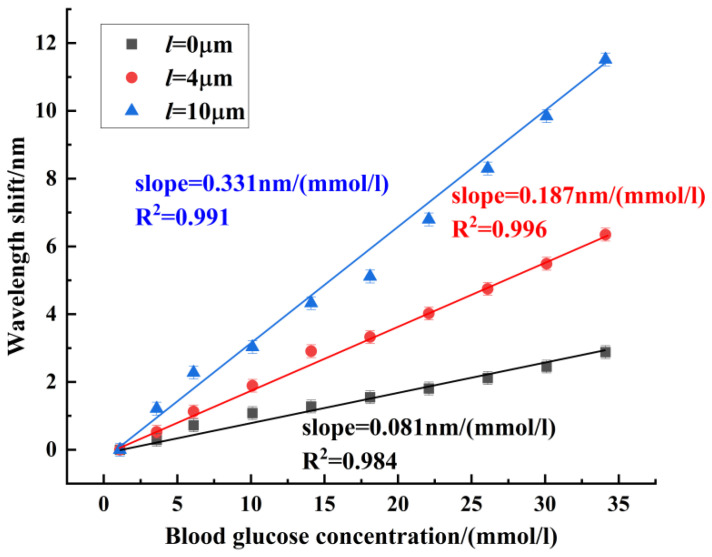
Experimental comparison of blood glucose concentration with different offset sensors.

**Figure 11 sensors-24-06344-f011:**
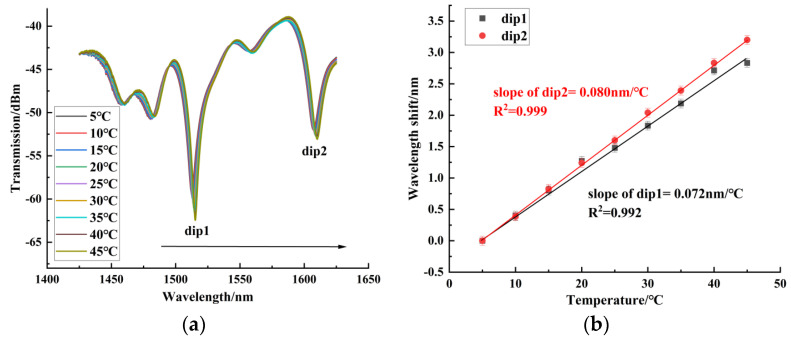
Temperature experimental results of sensor 1. (**a**) Spectrogram of the temperature experiment of sensor 1. (**b**) Experimental linear fit of temperature for Sensor 1.

**Figure 12 sensors-24-06344-f012:**
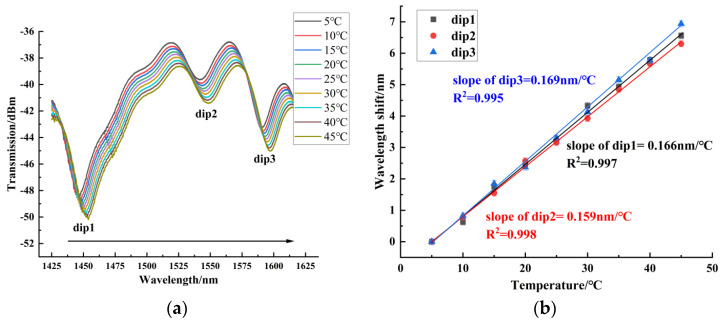
Temperature experimental results of sensor 2. (**a**) Spectrogram of the temperature experiment of sensor 2. (**b**) Experimental linear fit of temperature for Sensor 2.

**Figure 13 sensors-24-06344-f013:**
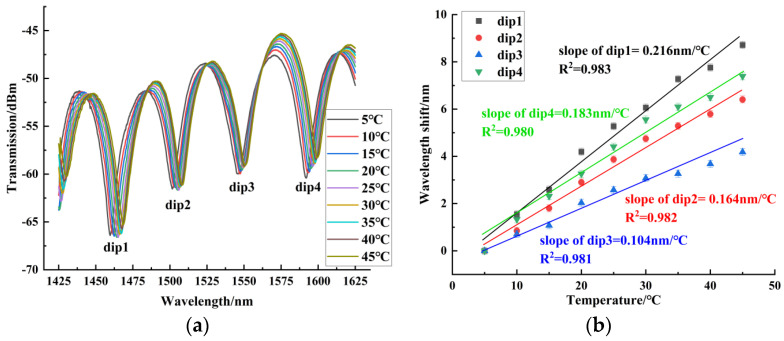
Temperature experimental results of sensor 3. (**a**) Spectrogram of the temperature experiment of sensor 3. (**b**) Experimental linear fit of temperature for Sensor 3.

**Figure 14 sensors-24-06344-f014:**
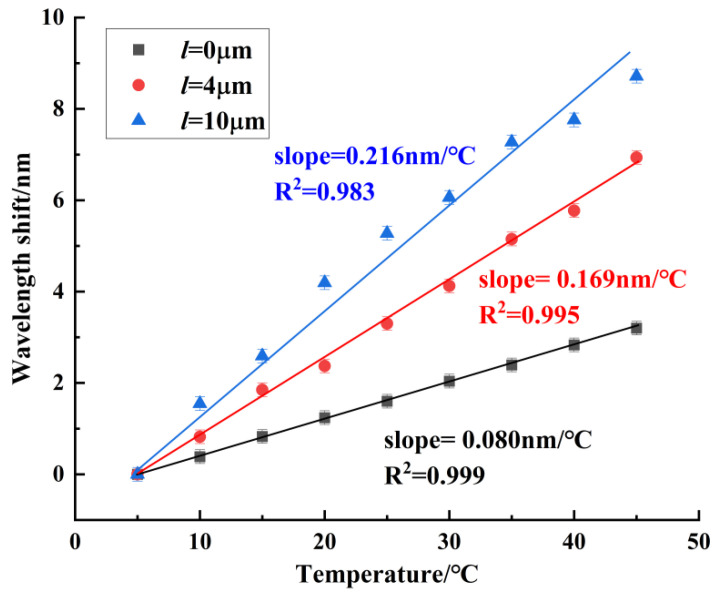
Experimental comparison of temperature with different offset sensors.

**Figure 15 sensors-24-06344-f015:**
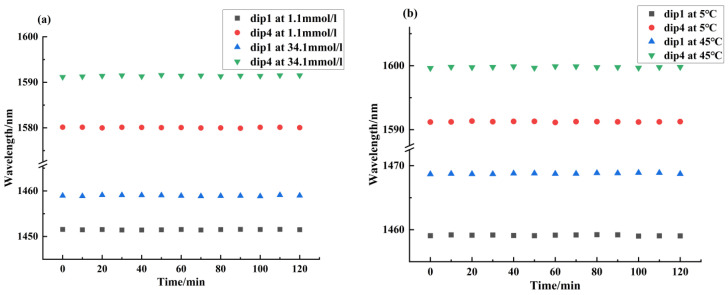
(**a**) Blood glucose concentration stability experiment results. (**b**) Temperature stability experiment results.

**Table 1 sensors-24-06344-t001:** Basic parameters of the three sensors.

Sensor	Bending Diameter (d)	Offset (l)
Sensor 1	1.5 cm	0 μm
Sensor 2	1.5 cm	4 μm
Sensor 3	1.5 cm	10 μm

**Table 2 sensors-24-06344-t002:** Comparison of different sensors.

Structure	Testing Time	Maximum Fluctuation
Glucose Oxidase-Modified U-Shaped SMF [6]	10 min	1.2 nm
No-core-fiber sensor based on surface plasmon [20]	5 times	0.715 nm
Surface plasmon resonance (SPR) sensor with Au film [21]	18 min	0.445 nm
This paper	120 min	0.17 nm

## Data Availability

The dataset generated and analyzed during this study is available from the corresponding author upon reasonable request, but restrictions apply to the commercially confident details.

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
