# Peer review of "Balloon-Shaped SMF Blood Glucose Concentration and Temperature Sensor Based on Core-Offset Structure"

_sensors, 2024, doi:10.3390/s24196344_

Round 1
Reviewer 1 Report
Comments and Suggestions for Authors
Please see the attached uploaded file

Comments on the Quality of English LanguageEnglish can be improved
Reviewer 2 Report
Comments and Suggestions for Authors
Comments:
In the research paper title “Balloon-shaped SMF blood glucose concentration and temperature sensor based on core-offset structure” results show that the maximum sensitivity of the sensors is 0.331 nm/(mmol/l) in the blood glucose concentration range from 1.1 mmol/l to 34.1 mmol/l, and the maximum sensitivity is 0.216 nm/℃ in the temperature range of 5 ℃ to 45 ℃. In addition, the sensors have characteristics such as simple structure, low cost, good stability, and electromagnetic interference resistance. Many literatures were reviewed and commented. This paper was favorable for promoting the development of efficient sensor. It is recommended to be published in Sensors after minor revision.
-Minor issues:
1. 0.331 nm/(mmol/l) shoul be 0.331 nM (mmol/l).
2. Redraw Figures 7 & 8.
3. The language and grammar in this paper needs polish.
-Major issues:
1. Author should be discussing the reason of “high linear fitting degree” for the maximum temperature sensitivity of sensors with different offsets. Means, why linear fitting occur and what is the scientific indication of linear fitting?
2. Author should be discussing about the red-shifted in this blood glucose concentration range. Why red shifted? What interaction going on between them?
Comments on the Quality of English LanguageNeed grammatical improvment.
Round 2
Reviewer 1 Report
Comments and Suggestions for Authors
The Authors have responded to the comments.
Comment: The table showing comparison of your sensor with the reported sensors should be added into the manuscript. Also, the Authors should include more references whether they have inferior or superior properties compared to your sensor.
Comments on the Quality of English LanguageEnglish has been improved
